# Homocysteine and Age-Related Central Nervous System Diseases: Role of Inflammation

**DOI:** 10.3390/ijms22126259

**Published:** 2021-06-10

**Authors:** Amany Tawfik, Nehal M. Elsherbiny, Yusra Zaidi, Pragya Rajpurohit

**Affiliations:** 1Department of Oral Biology and Diagnostic Sciences, Dental College of Georgia, Augusta University, Augusta, GA 30912, USA; drnehal@hotmail.com (N.M.E.); yzaidi@augusta.edu (Y.Z.); prajpurohit@augusta.edu (P.R.); 2James and Jean Culver Vision Discovery Institute, MCG, Augusta University, Augusta, GA 30912, USA; 3Department of Cellular Biology and Anatomy, Medical College of Georgia (MCG), Augusta University, Augusta, GA 30912, USA; 4Department of Ophthalmology, MCG, Augusta University, Augusta, GA 30912, USA; 5Eye Research Institue, Oakland University, Rochester, MI 48309, USA; 6Department of Biochemistry, Faculty of Pharmacy, Mansoura University, Mansoura 35516, Egypt

**Keywords:** hyperhomocysteinemia, Alzheimer’s disease, age-related macular degeneration, diabetic retinopathy, inflammation

## Abstract

Hyperhomocysteinemia (HHcy) is remarkably common among the aging population. The relation between HHcy and the development of neurodegenerative diseases, such as Alzheimer’s disease (AD) and eye diseases, and age-related macular degeneration (AMD) and diabetic retinopathy (DR) in elderly people, has been established. Disruption of the blood barrier function of the brain and retina is one of the most important underlying mechanisms associated with HHcy-induced neurodegenerative and retinal disorders. Impairment of the barrier function triggers inflammatory events that worsen disease pathology. Studies have shown that AD patients also suffer from visual impairments. As an extension of the central nervous system, the retina has been suggested as a prominent site of AD pathology. This review highlights inflammation as a possible underlying mechanism of HHcy-induced barrier dysfunction and neurovascular injury in aging diseases accompanied by HHcy, focusing on AD.

## 1. Introduction

Elevated levels of plasma homocysteine (Hcy), known as hyperhomocysteinemia (HHcy), is a major risk factor for neurodegenerative and cardiovascular disorders [1]. Over the last decade, elevated levels of amino acid Hcy have been frequently reported in patient with aging diseases. This relatively large incidence of HHcy in the elderly population is attributed to lowered nutritional absorption and decreased metabolic function with advanced age [2]. The percentage of the aging population has been increasing in the last 10 years and is expected to continue to grow for another 20 years, and this is attributed to improved life expectancies [3]. Among the most common 10 diseases affecting the aging population over the age of 65 are vision loss disabilities, such as diabetic retinopathy (DR) and age-related macular degeneration (AMD), neurodegenerative diseases, such as Alzheimer’s disease (AD), and osteoarthritis or osteoporosis [3], as shown in Figure 1.

Our and others’ work reported that HHcy causes disruption of the blood barrier function in both the brain [4,5,6,7] and retina [8,9,10,11,12,13,14]. The blood–brain barrier (BBB) separates the brain from the circulatory system and is made of tightly packed endothelial cells that line the cerebral vessels, separating blood stream components from the neuronal brain tissue [15]. This barrier is tightly maintained via specialized tight junctions, gap junctions, and adherent junctions and is vital for various features, such as: preventing entry of harmful substances to the neuronal tissue of the brain, performing selective transportation and trafficking of molecules into and out of the brain, and allowing specific ion transporter channels to regulate ionic transporters [16]. Similarly to the BBB, the blood–retinal barrier (BRB) regulates fluids and molecular movement between the ocular vascular and retinal tissues and prevents leakage of macromolecules and other potentially harmful agents into the retina. An intact BRB is vital for retinal structural and functional integrity as it plays an essential role in the maintenance of the retinal neuron microenvironment. The BRB consists of two components, an inner and outer BRB. Vision is negatively affected in clinical situations associated with BRB breakdown, such as DR, in which the inner BRB is altered [17,18,19], and AMD, in which the outer BRB is altered [18,19,20]. 

The current review emphasizes the involvement of HHcy-induced barrier dysfunction (BBB and BRB) in the development and progression of the most common vision loss diseases (DR and AMD) and the most common neurodegenerative (AD) disease in the elderly population and the possible underlying mechanisms that impair the barrier function. Our previously published work over the last few years has proposed many mechanisms for HHcy-induced barrier dysfunction, such as endoplasmic reticulum (ER) stress [21], activation of oxidative stress [11], induction of epigenetic modifications [9], induction of inflammation [22], and activation of a glutamate receptor, the N-methyl-D-aspartate receptor (NMDAR) [13]. 

## 2. Homocysteine and Alzheimer’s Disease

Alzheimer’s disease (AD) is the most common form of neurodegenerative disease and is the major cause of dementia, accounting for 60–70% of cases [23]. Worldwide, more than 26 million individuals have been diagnosed with AD. As the population ages, the prevalence of AD is expected to increase to effect over 100 million by 2050 [24]. Unfortunately, there is still no available effective treatment for AD, however, controlling the risk factors can still reduce the number of cases and associated cost, especially as AD is a devastating disease for the patients and their families and puts a huge financial burden on the whole of society [25]. The symptoms of this disease may start as simple symptoms, such as early forgetfulness, then deteriorate over time to gradual worsening in language, orientation, and behavior and late severe loss of memory and some bodily functions until eventual death [26]. The etiology of AD is complex and multi-factorial and still poorly understood. The main pathological features of AD are the intracellular accumulation of neurofibrillary tangles composed of hyperphosphorylated tau protein and increased production and deposition of amyloid-β (Aβ) with concomitant loss of synapses and neurons [27]. In addition, it is associated with BBB dysfunction [28,29]. AD progression has also been related to a gradual damage in function and structure in the hippocampus and neocortex areas of the brain involved in memory and cognition. These changes are concomitant with NMDAR activation and oxidative stress, ultimately resulting in AD pathology [30].

Hcy was elevated in patients with AD compared with normal controls and has been suggested as an independent risk factor for AD [31,32,33], to reduce the size and volume of the hippocampus and cortex of healthy elderly people [34], to sensitize hippocampal neurons to excitotoxins in animal models [35], to enhance neuronal death in mouse models of cerebral stroke [36], and to induce a dose-dependent increase in apoptotic cell death in cultured hippocampal neurons [35]. Indeed, HHcy was reported to double the risk of developing AD [37]. The underlying cellular mechanisms by which elevated Hcy induce neuronal death or exacerbate the consequences of other insults are still unclear. However, some potential mechanisms have been suggested for HHcy-induced brain damage and explain the connections between HHcy and AD, such as increasing cellular oxidative stress and hypo-methylation of DNA and proteins [38,39,40,41] ER stress [42], cerebrovascular damage [43], neuroinflammation [44], Aβ elevation [45,46], and tau protein phosphorylation [47]. 

One main connection between HHcy and AD pathology is the impairment of the BBB [48]. Elevated levels of Hcy were reported to compromise BBB integrity when the BBB was evaluated in a mouse model of HHcy [4,6], and was also reported to increase permeability of the BBB by NMDA receptor-dependent regulation of tight junctions [4]. Furthermore, microvascular disorders associated with HHcy were suggested as direct causal mechanisms linking vitamin B deficiency (B6, B12, and folic acid), resulting in HHcy and neurological dysfunction in AD [42]. In the early stages of AD, the impairment of BBB homeostasis induces the production of pro-inflammatory cytokines, which worsens synapse destruction and the accumulation and activation of microglia, while in the late stage of AD, amyloid deposits are frequently observed in larger blood vessels as well as smaller cerebral capillaries [49,50].

## 3. Homocysteine and Neurovascular Eye Diseases

Hcy was linked to many visual disorders. Over the last decade, Hcy was notably reported to be elevated in patients with retinal neurovascular diseases such as DR [51,52,53,54] and AMD [55,56,57,58]. Accumulating evidence from pervious publications linked HHcy to many vasculopathies, including endothelial dysfunction, vessel wall malformations, loss of extracellular matrix collagen, and disruption of the BRB in rodents and humans [59]. There is an association between HHcy and diabetes-induced microangiopathies (diabetic nephropathy, retinopathy, and macular edema) [60,61,62,63]. Furthermore, impaired endothelial cell function has been reported in HHcy both in vitro and in vivo [64]; however, the underlying cellular and molecular mechanisms have not yet been clearly defined. 

Our previous studies on a mouse model of HHcy caused by deficiency of the cystathionine-β-synthase enzyme (CBS) (cbs^−/−^ and cbs^+/−^ mice) reported an association between HHcy and retinal vascular dysfunction such as pathological neovascularization, central retinal vein occlusion, pericyte loss, and gliosis. Gliosis is a reactive change in glial cells in response to damage to the central nervous system (CNS) and involves the hypertrophy or proliferation of several different types of glial cells, including astrocytes [12] and microglia [22]. Besides the cbs (genetic) model of HHcy, we used another model of HHcy by injecting L-Homocysteine thiolactone hydrochloride locally in wild type (B57-BL6) mouse eyes (intravitreal injection) and we were able to confirm the changes we previously observed in the CBS mice. We reported retinal pigment epithelial (RPE) function disruption (barrier and phagocytic functions) by HHcy. We also reported the development of AMD-like features in mouse models of HHcy, and these features included vacuolization, hypopigmentation, atrophy, increased thickness of the basal laminar membrane, hyporeflective lucency, disturbed RPE–photoreceptor relationship, and development of choroidal neovascularization (CNV) [10]. 

## 4. Mechanisms of Homocysteine-Induced Neurovascular/Neurodegenerative Changes in the Central Nervous System (CNS)

The relation between elevated Hcy and neurovascular/neurodegenerative diseases has been established. Many mechanisms were proposed and studied for HHcy-induced vascular dysfunction, such as impaired endothelial function [65,66,67,68] oxidative stress [4,69,70], ER stress [71,72,73,74], inflammation [75,76,77,78,79], epigenetic modification [80,81,82,83], and activation of matrix metalloproteinase [84,85,86,87,88]. However, the exact mechanism of HHcy-induced vascular and barrier dysfunction is still undetermined.

Our work over the last decade has studied the effect of HHcy on retinal vasculature and reported that HHcy induced retina ischemia, tissue hypoxia, neovascularization, BRB dysfunction, and subsequent retinal hyperpermeability [8,10,12]. Moreover, this was followed by further studies to determine the underlying mechanisms of HHcy-induced vascular and BRB dysfunction. Given the anatomical fact that the retina is a part of the central nervous system, our studies on the BRB confirmed the effect of HHcy on the BBB. Remarkably, a growing body of evidence has demonstrated that AD affects both the brain and retina, with a significant correlation between pathological changes observed in both organs [89], highlighting the fact that retinal, neuronal, and microvascular alterations in AD could provide a unique window to the brain [90]. The retina is considered as an accessible “extension of the brain”, thereby it can be used as a non-invasive surrogate for detection and monitoring of AD-related pathological changes [91]. Based on these reports, we aimed to investigate if accumulation of β-amyloid and P-Tau, which are a characteristic pathological events of AD brains, occurs in the retina under HHcy conditions. Interestingly, our immunofluorescence data showed increased β-amyloid and P-Tau expression in RPE cells treated with Hcy (20 µM, 50 µM, 100 µM) in vitro and in primary RPE cells isolated from the genetic mouse model of HHcy (cbs) in vivo, as shown in Figure 2.

The current review will highlight the role of inflammation as a possible mechanism of HHcy-induced neurovascular changes and barrier dysfunction in aging diseases linked to HHcy, such as DR, AMD, and AD.

## 5. Homocysteine and Activation of Inflammation in Aging CNS Diseases

Accumulating evidence underscores the role of inflammation in the pathogenesis of aging diseases [92]. A strong association between HHcy and inflammation has been reported in various studies using human and experimental models [92,93,94]. Furthermore, HHcy was reported to show pro-inflammatory properties that led to visual dysfunction and optic nerve damage [51]. The association between HHcy and the induction of inflammation was attributed to two facts, first, the research that linked HHcy and the induction of inflammatory elements including leukocyte adhesion, expression of adhesion molecules, oxidative stress, reduced nitric oxide bioavailability, and endothelial dysfunction [95]. Additionally, another fact is that HHcy has been reported in association with various inflammatory diseases [96,97,98,99,100]. 

Given the acknowledged involvement of both HHcy and inflammation in age-related diseases, it is important to understand the role of HHcy in the induction of CNS inflammation. In this context, a recent study implies activation of Hcy/cytotoxic ceramide signaling as an underlying mechanism that propagates neuroinflmmation, degeneration, and apoptosis in AD [101]. Sun et al. reported an association between HHcy and AD development in elderly adults via potential mechanisms, including promoting inflammatory reactions, suppression of memory-related proteins, tau hyperphosphorylation, and Aβ accumulation [102]. Braun et al. found that HHcy altered inflammatory milieu, enhanced parenchymal plaque deposition, and triggered microglia function via upregulation of the expression of multiple “homeostatic” microglial genes in an experimental mouse model of AD [103]. In vitro, Hcy triggered a dose-dependent release of pro-inflammatory cytokines from adult astrocytes. Additionally, similar data was demonstrated in vivo, where an experimental rat model of mild HHcy showed inflammatory conditions in the cerebral cortex. Another suggested role for pro-inflammatory cytokines in Hcy-induced cognitive impairment is downstream activation of a pro-inflammatory-mediated increase in matrix metalloproteinase 9 (MMP9) which subsequently results in degradation of tight junctions, microhemorrhages, and, ultimately, cognitive impairment [104]. Further, auto-oxidation of Hcy, leading to cellular oxidative stress, could mediate HHcy-induced neurotoxicity through the formation of reactive oxygen species, causing neuroinflammation and apoptosis [105]. Additionally, accumulating evidence from other studies has reported that HHcy mediated vascular inflammation via activation of NF-κB in vascular smooth muscle cells [106] and contributed to the induction of cerebral ischemia via induction of cerebral microglia activation [107] and upregulation of pro-inflammatory cytokines [108]. 

Hcy and other pro-inflammatory factors such as interleukin-6 (IL-6), C-reactive protein (CRP), and alpha-1-antichymotrypsin (ACT) have been linked to neuroinflammation and cognitive decline [109]. Kommer et al. reported HHcy as an independent factor of lower level of information processing, general cognitive functioning, and fluid intelligence. Interestingly, the strongest negative correlation between HHcy and immediate recall was observed in persons with a high level of serum IL-6. HHcy was also negatively associated with retention in persons in the highest CRP tertile and with a faster rate of decline in persons in the lower and middle tertiles of CRP. Additionally, in the middle tertile of ACT, HHcy was associated with lower information processing speed and faster decline [110]. These results suggested that a combination of elevated Hcy and inflammation may be useful as a predictor of cognitive impairment. 

Various signaling pathways have been linked to Hcy-induced production of pro-inflammatory cytokines. Zou et al. reported that HHcy promoted DNA synthesis and activation of microglia via activation of the p38MAPK/NADPH oxidase/ROS pathway. Activated microglia in turn produce a diverse range of neurotoxic and pro-inflammatory factors [107]. Another possible link has been reported between HHcy and lowered cystathionine γ-lyase expression and H2S production in macrophages via triggering DNA hypermethylation [111]. Moroever, increased ROS and suppressed NO production by HHcy could explain the downstream inflammatory cascade [112]. This was evidenced by the ability of antioxidants such as N-acetyl cysteine and vitamin C and E to reduce Hcy-induced inflammation in animal models [113].

The disruption of the redox system in the aging population may be related to melatonin suppression. Increasing age is often associated with diminished endogenous melatonin production. Melatonin is a free radical scavenger that has shown the ability to protect the brain against various neurological injuries by direct free radical scavenging. Therefore, age-associated low melatonin production contributes to aggrevated Hcy-induced cerebral injury [114]. In this context, melatonin attenuated Hcy-induced cerebral lipid and protein oxidation in rat brains [115]. Another mechanism proposed for melatonin’s protective effect is the inhibition of apoptosis. Melatonin was found to inhibit Hcy-induced neural apoptosis by inhibiting mitochondrial cytochrome c release and restoring the anti-apoptotic/apoptotic protein balance. Additionally, melatonin inhibited Hcy-triggered DNA fragmentation by cleavage of poly(ADP-ribose) polymerase in hippocampal neurons of hyperhomocysteinemic rats [116]. Moreover, melatonin was able to modulate adhesion molecule expression in neural cells and improve learning and memory performances in hyperhomocysteinemic rats [117].

Consistent with these reports, we also reported the role of inflammation in aging retinal and brain diseases associated with HHcy such as DR, AMD, and AD [22]. Our recently published research showed that HHcy is accompanied by inflammation in both the retina and brain [22]. Indeed, mice genetically overexpressing Hcy demonstrated microglia activation and upregulated inflammatory markers in both retina and brain tissue. Moreover, similar results were obtained in vitro in Hcy-treated retinal pigment epithelial cells, human retinal endothelial cells, and monocyte cell lines. Analysis of supernatants from the aforementioned cells after treatment with Hcy indicated increased levels of pro-inflammatory cytokines along with downregulation of anti-inflammatory cytokines. Furthermore, nuclear translocation of transcription factor NF-κB that controls the expression of numerous inflammatory cytokines was also evaluated in vivo and in vitro. Interestingly, our data showed Hcy-induced NF-κB activation and nuclear translocation of NF-κB from the cytoplasmic to the nuclear compartments of cells under elevated Hcy conditions. These results suggested that HHcy induced inflammatory responses in the mouse brain, and retina and cultured retinal and microglial cells. Consequently, elimination of extra Hcy or prevention/attenuation of inflammation could be a promising intervention for alleviating damage associated with HHcy in age-related diseases such as DR, AMD, and AD.

Hcy acts as an agonist for metabotropic glutamate receptors as well as for NMDA receptors [118,119,120,121]. Considerable evidence supports the involvement of the NMDA receptor in the pathogenesis of AD [24,120,122,123,124]. Interestingly, its role in HHcy-induced neuronal degeneration [125,126] and BBB dysfunction was reported [4,127,128]. Our recent work highlighted its role in retinal vascular pathology and BRB dysfunction as well [13]. Interestingly, NMDA receptor activation has been linked to neuroinflammation and neurodegeneration involved in AD pathology [23]. On the other hand, inflammation, in addition to other factors, including oxidative stress, tau hyperphosphorylation, and Aβ deposition, has been associated with increased activity and/or sensitivity of the brain glutamatergic system, leading ultimately to neuronal dysfunction and cell death in AD [129]. It is believed that pro-inflammatory cytokines are released by Aβ-activated microglia. This in turn leads to disruption of redox and glutamate homeostasis and activation/sensitization of NMDA receptors, leading to neuronal cell death. Interestingly, NMDA receptor activation triggers Aβ production and deposition, leading to a vicious cycle. This viscous cascade could be blocked by memantine [130]. Gérard and Hansson reported that NMDA receptor activation is essential for triggering Ca^2+^ signaling and pro-inflammatory cytokine secretion in cultured astrocytes, contributing to reactive astrogliosis [131]. In brain microglia, activation of NMDA receptors triggered an inflammatory response and neuronal cell death, contributing significantly to cortical damage. This damage was markedly attenuated via pharmacological inhibition or genetically induced loss of NMDAR function in microglia [132]. All these data emphasize the crucial role of NMDA–inflammation crosstalk in the pathogenesis of AD.

Currently, there is no cure for AD. The Alzheimer’s Disease Medications Fact Sheet published by the National Institute on Aging shows few treatments options for AD. Mainly, two categories include the FDA-approved prescription drugs for the treatment of patients with AD, such as cholinesterase inhibitors, which are used for mild to moderate AD, and an inhibitor for NMDA receptors, memantine, for the treatment of moderate to severe AD. Memantine is an FDA-approved NMDAR antagonist, which possibly functions through suppressing extra-synaptic NMDAR signaling. Further studies are needed to help elucidate the molecular mechanisms of how glutamate and NMDARs function in the etiology of AD [23]. 

NMDA receptor activation has been related to synaptic dysfunction in AD. Synaptic NMDARs are neuroprotective and are required for the survival of neurons. However, over-activation of NMDARs located outside of the synapse play a key role in antagonizing the synaptic pro-survival signaling pathway by causing compression on the presynaptic and postsynaptic neurons and glial cells, loss of mitochondrial membrane potential, and cell death, which shift the balance toward excitotoxicity and neurodegeneration [24,31,58,133].

HHcy was reported to induce NDMDA receptor-dependent vascular inflammation, BBB disruption, and hippocampus synaptic dysfunction in mice. However, pharmacological inhibition of NMDA receptors mitigated theses effects [134,135]. Indeed, NMDA receptor stimulation mediates HHcy-induced oxidative injury in nerve terminals [136], neuronal cell death [35], and tau protein phosphorylation [137]. Therefore, Hcy–NMDA receptor stimulation may be the one key mechanism that may be fundamental for Hcy-induced neuronal damage in AD. Elevated Hcy levels in the brain overstimulate NMDA and α-amino-3-hydroxy-5-methyl-4-isoxazolepropionic acid receptors, resulting in higher production of free radicals, increased levels of cytoplasmic calcium, and activation of caspases, leading to cellular death. These events have also been reported to play a major role in BBB disruption through the increased activity of matrix metalloproteinases (MMPs) [138]. Interestingly, HHcy has been reported to induce CRP gene and protein expression, which in turn augmented NMDA receptor expression in rat vascular smooth muscles [139]. Therefore, Hcy can elicit a pro-inflammatory response in vascular smooth muscle cells of brain small arteries via stimulating CRP production accompanied by NMDA-ROS-MAPK-NF-κB signal pathway activation [113].

We believe that HHcy activates NMDARs and induces subsequent activation of many mechanisms, such as oxidative stress, ER stress, and inflammation. We also believe that there is a link between the changes occurring in aging eye diseases and AD disease with HHcy. Consequently, that led us to evaluate the eyes of mice with HHcy (CBS mice) for well-known markers of AD, such as beta amyloid (Aβ) protein and tau protein (T-tau), and tau phosphorylated at threonine 181 (P-tau181). Our data showed marked elevation of the AD marker in RPE cells by Hcy treatment in a dose-dependent manner, as shown in Figure 2. These two biomarkers have been incorporated into research diagnostic criteria for AD and have added importance in the differential diagnosis of AD and related conditions [140]. AD is characterized by altered levels of those makers in the cerebrospinal fluid (CSF) [141] and also in the plasma [142]. Various clnical and preclincal studies have linked HHcy to accumualtion of those markers in the brain of AD patients. Indeed, Hcy level was independently correlated with plasma Aβ 40 and 42 levels in AD patients [143]. Additionally, homocysteic acid (HA), an oxidized Hcy metabolite, induced accumulation of neurotoxic Aβ 42 in rat cortical neurons [144]. Further, Hcy increased vulnerability of vascular smooth muscle cells to Aβ, implying a role in increasing the risk of cerebral amyloid angiopathy, a morphological hallmark of AD [145]. This Hcy effect was explained by potentiation of c-secretase enzyme activity by an endoplasmic protein—Hcy-related protein (HERP)—that is formed in the presence of Hcy, promoting accumulation of Aβ proteins in the brain [43]. Another proposed mechanism is Hcy-induced upregulation of the presenilin 1 (PS1) gene—a key factor for Aβ formation in AD—by DNA hypomethylation. PS1 promotes amyloid precursor protein (APP) synthesis, triggering Aβ protein accumulation [146].

The crucial role of HHcy in AD pathology has been further evidenced by clinical studies that demonstrated the efficacy of B vitamin supplementation on AD development, as deficiency of these vitamins causes Hcy accumulation in the body. A randomized controlled trial showed that folic acid supplementation improved cognition and mitigated inflammation in AD patients [147]. A high intake of folic acid has also been linked to a decrease in the risk of AD development in elderly people [148]. Moreover, a recent cohort study concluded that inadequate vitamin B12 intake intensified cognitive decline in patients with dementia [149]. Another randomized control trial demonstrated the clinical benefits of B vitamin supplementation for cognitive decline in AD patients (folic acid, B6, B12) [150].

In conclusion, it is obvious that both brain and retinal tissue are subjected to a linked cascade of events during pathogenesis of AD. HHcy-induced inflammation plays a crucial role during these events, resulting ultimately in neurodegeneration, barrier dysfunction, and cognitive and retinal impairment. Studying common pathogenic features between the brain and retina during AD development could lead to better understanding of HHcy’s role in different cellular mechanisms, including inflammation, and hence the accelerated development of therapeutic interventions that target Hcy itself or its downstream key signaling pathways.

## Figures and Tables

**Figure 1 ijms-22-06259-f001:**
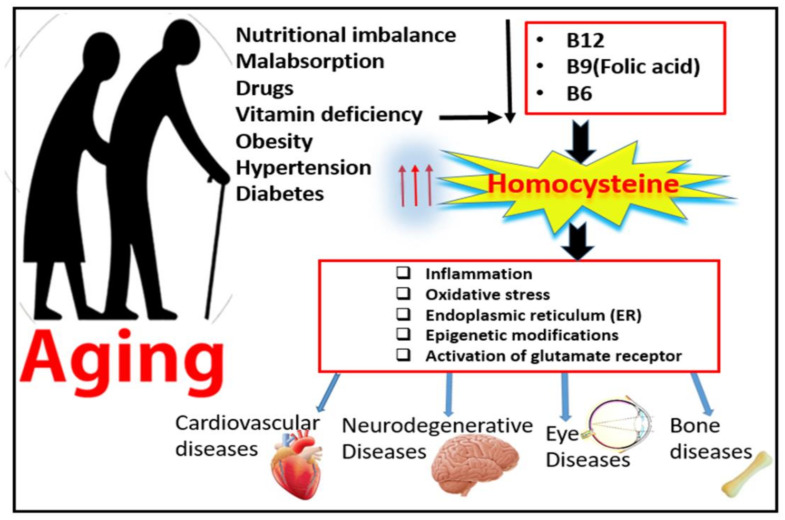
Effect of aging on homocysteine metabolism and possible mechanisms of hyperhomocysteinemia-associated aging diseases.

**Figure 2 ijms-22-06259-f002:**
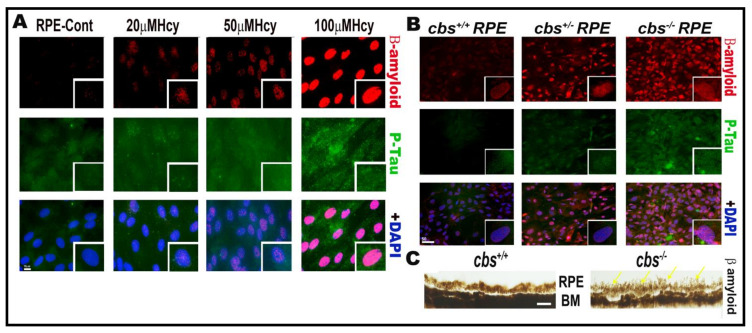
Hcy induces accumulation of β-amyloid and P-Tau proteins in retinal RPE. IF staining showed increased expression of β-amyloid (red) and P-Tau (green) in (**A**) RPE treated with and without Hcy (20 μM, 50 μM, and 100 μM). (**B**) Primary RPE cells isolated from cbs^+/−^ and cbs^−/−^ mouse retina. (**C**) Histological staining of retinal sections with thioflavin S stain showing more accumulation of β-amyloid protein in RPE area of the cbs^−/−^ mouse retina. The cbs^+/−^ mice (heterozygous) represent mild/moderate HHcy, have about a 4- to 7-fold increase in plasma Hcy level, and show a mild retinal phenotype and normal life span, while the cbs^−/−^ mice (homozygous with no copies of cbs) represent severe HHcy, have about a 30-fold increase in plasma Hcy with severe retinal phenotype and a short life span of ~3 to 5 weeks. Scale bars: 50 μm.

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
