# Peer review of "Homocysteine and Age-Related Central Nervous System Diseases: Role of Inflammation"

_ijms, 2021, doi:10.3390/ijms22126259_

Round 1

Reviewer 1 Report

This manuscript presents an established concept that has been described by many authors (recently by Moretti & Caruso). To make this manuscript more valuable, the authors should discuss several questions. The authors say that the aim of review is to highlight the role of inflammation in aging diseases. But they repeatedly say that effects of HHcy may due to induced impaired endothelial function stress, ER stress, inflammation, epigenetic modifica tion and activation of matrix metalloproteinase (line 144, 189, etc…), but the mechanism by Homocysteine and inflammation may be predictors of cognitive decline in older persons is not discussed, as well as signal pathways underlying homocysteine-induced production of cytokines

The manuscript is not well balanced, in fact an extensive description of the NMDA receptors, but few information about the correlation between Hcy increase and Abeta 1–40 deposition in the brain of AD patients [Irizarry, M.C.; Gurol, M.E.; Raju, S. Association of homocysteine with plasma amyloid beta protein in agingand neurodegenerative disease.Neurology2005,65, 1402–1408.], and induction and potentiation by Hcy of the intracellular and extracellular accumulation of Abeta 42 [Hasegawa, T.; Ukai, W.; Jo, D.-G. Homocysteic acid induces intraneuronal accumulation of neurotoxicAbeta42, implications for the pathogenesis of Alzheimer’s disease.J. Neurosci. Res.2005,80, 869–876.], or the ability of HHcy that, by DNA hypomethylation can lead to up-regulation of presenilin genes,in particular, the one regulating presenilin 1 (PS1). The HHcy induction up-regulates PS1 gene, and therefore increases APP, promoting, therefore, the amyloid cascade sequence

In addition, Hcy may be linked to lower melatonin production. In fact melatonin scavenges free radicals and counteracts Hcy by a direct antioxidant effect and by apoptosis modulation, but no discussion of this is present in the manuscript

Also informations on the results of clinical studies on the efficacy of folic therapy and vitamin B12 combined in reducing the accumulation of homocysteine, could improve the manuscript.

There are substantial shortcomings in terms of the wording and clarification of acronyms throughout the manuscript, which give the impression that it was not carefully written and corrected prior to submission.

This impression of sloppiness distracts from the overall significance and comprehensiveness of the review.

Author Response

  • This manuscript presents an established concept that has been described by many authors (recently by Moretti & Caruso). To make this manuscript more valuable, the authors should discuss several questions. The authors say that the aim of review is to highlight the role of inflammation in aging diseases. But they repeatedly say that effects of HHcy may due to induced impaired endothelial function stress, ER stress, inflammation, epigenetic modifica tion and activation of matrix metalloproteinase (line 144, 189, etc…), but the mechanism by Homocysteine and inflammation may be predictors of cognitive decline in older persons is not discussed, as well as signal pathways underlying homocysteine-induced production of cytokines

Thank you for your comment. A paragraph was added (Lines 196-203) to describe the mechanism by which Homocysteine and inflammation may be predictors of cognitive decline in older persons.  Another paragraph was added to discuss signal pathways underlying homocysteine-induced production of cytokines (Lines 204-210).

  • The manuscript is not well balanced, in fact an extensive description of the NMDA receptors, but few information about the correlation between Hcy increase and Abeta 1–40 deposition in the brain of AD patients [Irizarry, M.C.; Gurol, M.E.; Raju, S. Association of homocysteine with plasma amyloid beta protein in agingand neurodegenerative disease.Neurology2005,65, 1402–1408.], and induction and potentiation by Hcy of the intracellular and extracellular accumulation of Abeta 42 [Hasegawa, T.; Ukai, W.; Jo, D.-G. Homocysteic acid induces intraneuronal accumulation of neurotoxicAbeta42, implications for the pathogenesis of Alzheimer’s disease.J. Neurosci. Res.2005,80, 869–876.], or the ability of HHcy that, by DNA hypomethylation can lead to up-regulation of presenilin genes,in particular, the one regulating presenilin 1 (PS1). The HHcy induction up-regulates PS1 gene, and therefore increases APP, promoting, therefore, the amyloid cascade sequence.

Thank you for your comment. We summarized the part regarding NMDA a paragraph and discussion of role of Hcy in deposition of amyloid beta proteins in AD and underlying mechanisms using the suggested references was added (Lines 288-296).

  • In addition, Hcy may be linked to lower melatonin production. In fact, melatonin scavenges free radicals and counteracts Hcy by a direct antioxidant effect and by apoptosis modulation, but no discussion of this is present in the manuscript

Thank you for your suggestion. The link between Hcy-induced injury and melatonin in elderly people was described (Lines 211-220).

  • Also, information on the results of clinical studies on the efficacy of folic therapy and vitamin B12 combined in reducing the accumulation of homocysteine, could improve the manuscript.

Results of clinical studies on the efficacy of B vitamins supplementation on HHcy-induced cognitive decline were cited (Lines 297-303).

  • There are substantial shortcomings in terms of the wording and clarification of acronyms throughout the manuscript, which give the impression that it was not carefully written and corrected prior to submission. This impression of sloppiness distracts from the overall significance and comprehensiveness of the review.

Thank you for your comment. The review was thoroughly revised, typo errors were corrected, and acronyms were clarified.

Reviewer 2 Report

The review suffers from a few problems, mainly in the inappropriate citation of some of the references and in the use of the English language. Matters needing attention are:

line 14: replace 'reported' by 'common'

16: add 'AD' as an abbreviation

29: homocysteine is mis-spelt

30: reference to a recent review would be helpful here, such as Smith AD, Refsum H. Homocysteine - from disease biomarker to disease prevention. J Intern Med. 2021. doi: 10.1111/joim.13279

31: replace 'remarkably' by 'frequently'

34: the reference 1 cited here is not appropriate for this statement

37 and 39: remove he semicolons

45: Figure 1 is purely descriptive and is not informative - it shoudl be removed

49: remove 'brain' from 'cerebral brain'

82: reference 23 is totally inappropriate

91: replace 'reported' by 'elevated'

92: a more recent review could be cited, such as the International Consensus Statement: Smith AD, Refsum H, Bottiglieri T, Fenech M, Hooshmand B, McCaddon A, et al. Homocysteine and dementia: An international consensus statement. J Alzheimers Dis. 2018; 62: 561-70.

97: the reference 35 is not appropriate. Cite a meta-analysis such as Beydoun MA, Beydoun HA, Gamaldo AA, Teel A, Zonderman AB, Wang Y. Epidemiologic studies of modifiable factors associated with cognition and dementia: systematic review and meta-analysis. BMC Public Health. 2014; 14: 643.

110: a more appropriate reference than 30 would be:   Troen AM, Shea-Budgell M, Shukitt-Hale B, Smith DE, Selhub J, Rosenberg IH. B-vitamin deficiency causes hyperhomocysteinemia and vascular cognitive impairment in mice. Proc Natl Acad Sci U S A. 2008.

117: patient shoud be patients

187: reference 95 is totally inapprpriate.

212: replace 'preceding' by 'other'

217: remove 'In' from 'In consistent'

241: what does a paper on colchicine have to do with AD? Replace ref. 120

245: reference 121 deals with melanoma and is totally inappropriate

257-264: this paragraph adds nothig new and can be removed

266: remove 'recently'

272-274: this sentence can be omitted as it adds nothing

295-296: this statement is WRONG. beta-amyloid levels are not increased, but decreased, in CSF of patients with AD, as the reference cited shows.

Author Response

The review suffers from a few problems, mainly in the inappropriate citation of some of the references and in the use of the English language. Matters needing attention are:

  • line 14: replace 'reported' by 'common.'

“reported” was replaced by “common”, thank you.

  • 16: add 'AD' as an abbreviation.

AD was added, thank you.

  • 29: homocysteine is mis-spelt.

We apologize for this typo mistake; the spelling was corrected.

  • 30: reference to a recent review would be helpful here, such as Smith AD, Refsum H. Homocysteine - from disease biomarker to disease prevention. J Intern Med. 2021. doi: 10.1111/joim.13279

Thank you for your suggestion. The suggested reference was added.

  • 31: replace 'remarkably' by 'frequently.'

Remarkably was replaced by frequently, thank you.

  • 34: the reference 1 cited here is not appropriate for this statement.

Thank you for your comment.  The sentence was rephrased, and the reference was replaced by an appropriate one.

  • 37 and 39: remove he semicolons.

Thank you for your comment. The semicolons were removed.

  • 45: Figure 1 is purely descriptive and is not informative - it shoudl be removed.

Thank you very much and we really appreciate your valuable comment regarding figure 1 and we agree it was poorly descriptive.  We aimed to summarize what was discussed in the review regarding homocysteine metabolism, development of hyperhomocysteinemia, and its link to aging disease. We appreciate your comments which helped us to improve the figure by adding the possible mechanisms that were discussed in the current review as we think that this figure will help the readers to follow the review smoothly.  

  • 49: remove 'brain' from 'cerebral brain.'

“Brain was removed.”

  • 82: reference 23 is totally inappropriate.

Thank you for your comment. the reference was replaced by appropriate one.

  • 91: replace 'reported' by 'elevated.'

“Reported” was replaced by “elevated.” 

  • 92: a more recent review could be cited, such as the International Consensus Statement: Smith AD, Refsum H, Bottiglieri T, Fenech M, Hooshmand B, McCaddon A, et al. Homocysteine and dementia: An international consensus statement. J Alzheimers Dis. 2018; 62: 561-70.

Thank you for your comment. The suggested reference was added.

  • 97: the reference 35 is not appropriate. Cite a meta-analysis such as Beydoun MA, Beydoun HA, Gamaldo AA, Teel A, Zonderman AB, Wang Y. Epidemiologic studies of modifiable factors associated with cognition and dementia: systematic review and meta-analysis. BMC Public Health. 2014; 14: 643.

Thank you for your comment. The suggested reference was cited.

  • 110: a more appropriate reference than 30 would be:   Troen AM, Shea-Budgell M, Shukitt-Hale B, Smith DE, Selhub J, Rosenberg IH. B-vitamin deficiency causes hyperhomocysteinemia and vascular cognitive impairment in mice. Proc Natl Acad Sci U S A. 2008.

Thank you for your comment. The suggested reference was cited.

  • 117: patient shoud be patients.

Patients was changed to patients.

  • 187: reference 95 is totally inapprpriate.

Thank you for your comment. Reference was changed to appropriate one.

  • 212: replace 'preceding' by 'other.'

'preceding' was replaced by 'other.'

  • 217: remove 'In' from 'In consistent.'

In was removed

  • 241: what does a paper on colchicine have to do with AD? Replace ref. 120.

Thank you for your comment. The reference was replaced.

  • 245: reference 121 deals with melanoma and is totally inappropriate.

Thank you for your comment. The reference was replaced.

  • 257-264: this paragraph adds nothig new and can be removed.

The paragraph was removed.

  • 266: remove 'recently',

'recently' was removed.

  • 272-274: this sentence can be omitted as it adds nothing.

The sentence was removed.

  • 295-296: this statement is WRONG. beta-amyloid levels are not increased, but decreased, in CSF of patients with AD, as the reference cited shows.

Thank you for your careful comment. The sentence was rephrased to align with the reference.

Round 2

Reviewer 2 Report

The revisions have covered my concerns